# Ultrasmall-in-Nano: Why Size Matters

**DOI:** 10.3390/nano12142476

**Published:** 2022-07-19

**Authors:** Ryan D. Mellor, Ijeoma F. Uchegbu

**Affiliations:** School of Pharmacy, University College London (UCL), 29–39 Brunswick Square, London WC1N 1AX, UK; ryan.mellor.16@ucl.ac.uk

**Keywords:** ultrasmall-in-nano, gold nanoparticles, clearance, biodistribution, tumor accumulation, toxicity

## Abstract

Gold nanoparticles (AuNPs) are continuing to gain popularity in the field of nanotechnology. New methods are continuously being developed to tune the particles’ physicochemical properties, resulting in control over their biological fate and applicability to in vivo diagnostics and therapy. This review focuses on the effects of varying particle size on optical properties, opsonization, cellular internalization, renal clearance, biodistribution, tumor accumulation, and toxicity. We review the common methods of synthesizing ultrasmall AuNPs, as well as the emerging constructs termed ultrasmall-in-nano—an approach which promises to provide the desirable properties from both ends of the AuNP size range. We review the various applications and outcomes of ultrasmall-in-nano constructs in vitro and in vivo.

## 1. Introduction

Colloidal gold is the subject of ever-growing interest in the field of nanotechnology. This is due to its versatility and tunability in terms of size, shape, and surface chemistry. With a rigorous understanding of the properties of gold nanoparticles (AuNPs) comes the ability to exploit them for a plethora of therapeutic and diagnostic applications.

Almost any material will display three distinct size-dependent ranges of properties, in their atomic-, nano-, and bulk-scale [1]. Thus, most materials can feasibly exist as a ‘nanomaterial’ between 1 and 1000 nanometers; however, to be of any practical use, its properties must be precisely and reproducibly manipulated at scale and, under this criteria, AuNPs excel. Various methods (chemical and physical) have been developed to accurately control AuNP’s size (from 1 to 330 nm), shape (spheres, rods, stars, plates, cubes, cages, and shells), surface chemistry, and optical-electronic properties. Furthermore, AuNPs are inert, non-toxic, and can be made to be stable in a range of solvents and pH values, properties which are desirable from a biological standpoint [2].

With progress into the tunability of almost every aspect of AuNPs comes the opportunity to investigate how varying each one of these properties independently will affect the physicochemical properties and biological outcome of the particles. One of the easiest and most effective ways to control the properties of AuNPs is by varying the size. There are advantages and disadvantages for AuNPs in both the ultrasmall (<5 nm) and the nano (5–1000 nm) size range, in terms of the optical properties, cellular uptake, opsonization, toxicity, biodistribution, tumor accumulation, and excretability. This work will outline some of the methods implemented to synthesize ultrasmall AuNPs, the trends observed with varying AuNP size, and finally, the approaches and applications of ultrasmall-in-nano—a new construct which is able to combine the advantages from both ends of the size range.

## 2. Effects of Varying Size

### 2.1. Effect on Optical Properties

Varying the size of AuNPs is one of the most straightforward and efficient ways to alter the optical properties and enhance scattering [3]. AuNPs exhibit surface plasmon resonance (SPR), meaning that, at a particular range of wavelengths, dependent on the particles’ size and shape, they will display increased absorbance. This is a distinctly nano property, not observed at either the atomic or bulk scale of gold [4]. This SPR band can be exploited to create diagnostic techniques such as surfaced enhanced Raman spectroscopy (SERS) [5] and spatially offset Raman spectroscopy (SORS) [6], and for therapeutic techniques including photothermal therapy (PTT) [7] and photodynamic therapy (PDT) [8]. For diagnostic and therapeutic applications, the SPR band should reside in a region known as the phototherapeutic window—650–850 nm, a range of low absorbance by biomolecules in human tissue, resulting in the high depth penetration of the incident laser light [9]. Unfortunately, particles which exhibit this SPR band tend to be in the size range of 100–200 nm [10], a size that would not be sufficiently excreted by the kidneys.

Generally, larger particles will have a more redshifted SPR peak compared to their smaller counterparts [11]. Spheres in the ultrasmall size range (~5 nm) will exhibit SPR peaks in the range of 515–520 nm and this will bathochromically shift to over 570 nm with an increasing particle diameter beyond 100 nm [12]. This bathochromic shift is accompanied by a broadening of the resonance [3,11].

It is also reported that SERS intensity increases with particle size [11], this is of huge significance for theranostic applications of AuNPs as a stronger SERS intensity results in particles which can be detected at lower concentrations and greater depth. Further investigation is required to determine the interplay between particle size, laser wavelength, and SERS intensity [11].

### 2.2. Effect on Opsonization

Nanoparticles, upon introduction to the bloodstream, collide with proteins, some of which will bind to the particle, forming a biological identity known as a protein corona. A fraction of these proteins will be opsonins, such as complement proteins [13] and antibodies [14], which tag the particle for uptake by phagocytes and elimination from the body. This is a vital aspect of the immune system and it is responsible for the removal of, among other things, pathogens, diseased cells and protein aggregates [15]; however, this phagocytosis leads to the undesirable reduced circulation time of nanoparticles. One of the most successful approaches to avoid opsonization is by altering the nanoparticle’s surface chemistry, for example, by PEGylation [16,17]; however, the particle’s size can also have a dramatic effect.

It has been shown that, for nanoparticles and proteins of similar sizes, their attraction resulting from van der Walls potential scales with the particles radii [18]. Therefore, it follows that larger particles will experience a greater degree of opsonization, as has been shown experimentally for particles with diameters of 7–22 nm [19]. The increased opsonization can be explained in part by the increased surface area; however, this is not a complete explanation, as it has been shown that the density of adsorbed proteins also increases [20]. This trend would not be expected to hold indefinitely, since there reaches a point where, due to the decreasing curvature, the particle’s surface is indistinguishable from being flat from the perspective of a protein. Indeed, that size limit has been shown to be on the order of 50 nm, after which the protein binding diminishes with increasing size [21].

### 2.3. Effect on Cellular Internalization

It is usually desirable to either suppress or promote cellular internalization, depending on the theranostic application. Suppression is generally advantageous if the application does not require activity within the cell, for example, imaging or PTT; this allows efficient elimination of the nanoparticle after it has served its purpose [22]. Conversely, the promotion of cellular internalization also has its applications, for example, the delivery of cargo which acts intracellularly. However, when a system is designed to promote initialization, efforts should be made to minimize systemic toxicity arising from non-specific internalization, and this may be achieved by coating the particle with a targeting moiety [23].

Nanoparticles rely heavily on receptor-mediated endocytosis for cellular internalization [24,25]. This results in the rate of internalization, with regards to nanoparticle size, being governed by two opposing phenomena. Larger particles can bind to many cell-surface receptors simultaneously, which leads a decrease in the Gibbs free energy, resulting in the membrane wrapping around the particle. On the one hand, a smaller particle is only able to interact with a few receptors at a time; this means that there is no risk of a localized receptor shortage which would act to reduce the overall rate of particle uptake. Similarly, larger particles require a greater surface area of the cell membrane to envelope the particle, whereas smaller particles require a greater surface curvature of the cell membrane, neither of which are favorable due to either kinetics or energetics, respectively. Several studies, both theoretical [25,26,27] and experimental [28,29], have found the sweet spot for efficient internalization to be around 25–50 nm. However, this range is heavily dependent on the particles’ surface chemistry [30] and cell-line [31] being tested. More generally, 10–100 nm is considered the optimal range for favoring internalization, where particles outside of this range will suffer from one of the aforementioned phenomena.

### 2.4. Effect on Renal Clearance

The renal clearance of injected agents is desirable to avoid hazards resulting from the accumulation and/or decomposition of said agents. In order for efficient renal clearance, particles must poses a hydrodynamic diameter below that of the kidney filtration threshold (KFT) of ~5.5 nm [32,33], a limit set by the size of the glomerular pores that filter blood plasma. Particles of up to 8 nm may be filtered by the kidneys provided so that the particle surface is positively charged; this is due to favorable interactions with the filtration barrier, which are not present for neutral or negatively charged particles [34]. Particles that are not able to be filtered by the kidneys will rely on the slower hepatobiliary pathway for clearance, if they are to be excreted at all.

It is noteworthy that there exists a Goldilocks zone for the size of renally clearable particles of approximately 1–5 nm; this is assumed to be due to fact that particles below 1 nm can enter the ~1 nm pores of the glomerular glycocalyx. This is evidenced by the exponential decrease in rate of glomerular filtration with increasing particle size for atomically precise AuNPs below 1 nm, where the size is inferred from the particles’ mass as determined by electrospray ionization mass spectroscopy [35].

Multiple studies [36,37] have compared particles within the size range 1–5 nm to particles larger than 5 nm and the findings are in almost unanimous agreement. That is, larger (>5 nm) particles show low to undetectable levels in the urine as they cannot pass through the glomerular pores, and significant levels accumulating in the liver and excreted via the hepatobiliary pathway. Meanwhile, smaller (<5 nm) particles show higher rates of excretion via both the renal and hepatobiliary pathways, where more than 50% of the injected dose (%ID) is cleared in hours [38], rather than weeks [39] to months [40], for larger particles. Notably, it is the hydrodynamic diameter, and not that of the solid core, which determines the fate of a nanoparticle. This is nicely demonstrated by varying the molecular weight, and therefore the thickness, of a PEG coating while maintaining a gold core size of 2.5 nm; the <5 nm hydrodynamic diameter particles showed preferential renal clearance, while the >5 nm hydrodynamic diameter particles showed a decreased rate that would be expected of larger particles [41]. The correlation of size vs. rate of excretion has been shown to be exponential in nature [42], where particles of 2, 6, and 13 nm demonstrated renal clearance efficiencies of 50, 4, and 0.5%, respectively, 24 h post-injection.

### 2.5. Effect on Biodistribution

The biodistribution of nanoparticles is heavily impacted by not only the particle size, but also the shape and surface chemistry; therefore, it is important to bear in mind that the trends observed when changing any one of these parameters can be enhanced or countered by changing one of the others. That said, all else being equal, there are significant correlations between biodistribution and size. Figure 1 summarizes some of the ways in which biodistribution is affected by the particles’ properties.

Generally, it is shown that smaller particles exhibit a more widespread distribution than larger ones. One study compared the biodistribution of 10, 50, 100, and 250 nm particles 24 h after intravenous injection [43]. They found detectable levels of the largest particles (100 and 250 nm) in the blood, liver, and spleen, with negligible quantities (0.1 %ID) in the kidneys; smaller 50 nm particles were further detected in the lungs and heart; and only the smallest 10 nm particles were detected in the remaining tissues in the testis, thymus, and brain, with a larger quantity (1 %ID) present in the kidneys. Across all sizes tested, the highest organ accumulation was found to be in the liver. A similar study, comparing 15, 50, 100, and 200 nm particles, found almost identical results, with the exception that 50 nm particles were additionally detected in the brain [44]; this may be attributed to the different animal models and AuNP preparations used in the two studies [45].

### 2.6. Effect on Tumor Accumulation

For any cancer theranostic, it is necessary to achieve some degree of targeting. In the case of diagnostic applications, a construct must be able to accumulate at the tumor in order to distinguish it from healthy tissue and identify the tumor. For therapy, tumor accumulation allows tissue damage to be focused on the diseased tissue to the greatest extent possible, maximizing efficacy and minimizing adverse side effects.

One popular approach is to functionalize the surface of constructs with tumor-targeting antibodies, aptamers, peptides, or small molecules; however, this is outside of the scope of this review; we refer the reader to the review “Active targeting of gold nanoparticles as cancer therapeutics” [46] for an overview of the subject. Within the scope is the opportunity for size-based passive targeting. Nanoparticles need to fall within a particular size rage (40–400 nm [47]) in order to take advantage of the enhanced permeability and retention (EPR) effect. The EPR effect leads to passive preferential accumulation in tumor tissue due to a combination of the leaky vasculature, compared to the continuous endothelial junctions of healthy tissue, and reduced lymphatic drainage that would normally help to carry away cytotoxic compounds [48]. For passive targeting to be effective, the therapeutic must remain in circulation for as long as possible. To accomplish an extended half-life, the NPs may be functionalized to tune the particle’s size, surface charge, hydrophobicity, and surface chemistry in order to reduce the renal and phagocytic clearance [49].

Size plays a key role in a particle’s ability to preferentially accumulate at a tumor. Small particles show more widespread distribution amongst all tissues, both healthy and diseased; they can diffuse more freely into the tumor from the vasculature [50], but this also means that they diffuse out again at a greater rate. Larger particles, close to or exceeding 1 µm, are unable to pass through the tumor fenestrae, with pores in the order of a few hundred nanometers [51]; they are also less able to diffuse into solid tumors, which may not be problematic if the intended application is diagnostic, where locating at the tumor boundary is sufficient.

As discussed in the previous section, smaller particles exhibit extended circulation times and this tends to lead to increased tumor accumulation compared to larger particles; ultrasmall AuNPs have been shown to have a particularly high tumor accumulation when compared to >10 nm AuNPs [52].

### 2.7. Effect on Toxicity

The correlation between size and toxicity of AuNPs is not easy to discern from the literature, due to inconsistent synthesis methods, capping ligands, cell/animal models, dosages, and routes of administration [53]. Toxicity has been reported for ultrasmall (<2 nm) AuNPs, and when the bare gold surface is accessible [54]. The toxicity of ultrasmall AuNPs may be attributed to the more widespread biodistribution and longer circulation times when compared to larger particles which rapidly accumulate in the liver.

Most studies report AuNPs as being non-toxic, and where toxicity is observed, it is generally attributed to physicochemical properties derived from the capping ligand as opposed to the gold core itself.

On the other hand, one study looked at citrate-capped particles ranging from 3 to 100 nm, performing an in vitro MTT ((3-(4,5-dimethylthiazol-2-yl)-2,5-diphenyltetrazolium bromide) assay against Hela cells to assess the particles and found no cytotoxicity for any size at any concentration (up to 0.4 mM). The study also determined the average lifespan (L_50_) of BALB/c mice dosed intraperitoneally with 8 mg/kg/week for each particle size and no toxicity or lethality was observed for small particles (3, and 5 nm) or for large particles (50, and 100 nm); however, intermediate particles (8, 12, 17, and 37 nm) all showed an L_50_ of less than 21 days. The authors suggest that the zone of toxicity is attributed to particles being small enough to enter cells but large enough to avoid initiating a specific immune response [55].

## 3. Methods to Synthesize Ultrasmall AuNPs

A plethora of methods have been developed to precisely control the size, shape, and surface chemistry of AuNPs. These range from green synthesis methods, where the AuNPs are produced either by microorganisms or plant extracts [56,57,58], to physical methods such as laser ablation [59,60,61], thermal decomposition [62,63], and mechanical milling [64], and finally, chemical synthesis methods. Chemical synthesis methods are perhaps the most widely employed methods, owing to the vast array of physicochemical properties which can be achieved and the specificity with which they can be obtained. Many approaches exist to chemically synthesize AuNPs; however, they all proceed via essentially the same steps (Figure 2):**Reduction** of Au^3+^—afforded by a gold salt, usually HAuCl_4_—to atomic Au^0^; this process is rapid and continues until the concentration of gold atoms in solution reaches supersaturation.**Nucleation** of gold atoms into gold clusters; the number of nucleation sites determines the number concentration of AuNPs, i.e., for a fixed mass concentration more nucleation events results in smaller particles and vice versa.**Growth** via coalescence of gold clusters and diffusion of remaining soluble gold atoms onto the surface of gold agglomerates.

The following sections will outline some of the most common methods used to synthesize ultrasmall AuNPs; they have been divided into the four main methods used in the literature—Turkevich/Frens, reduction by sodium borohydride, Brust–Schiffrin, and seeded growth. Where reagent names have been abbreviated, the meaning can be found in Table 1.

**Turkevich/Frens** (Figure 3A) synthesis is the classical method of producing AuNPs. It was one of the first systematic approaches to the size-controlled synthesis of AuNPs and is still popular today, owing largely to its simplicity and reliability. The method was pioneered Turkevich et al. in 1951 [65], producing 15–24 nm AuNPs, and later refined by Frens in 1973 [66], extending the size range to 16–147 nm. In this synthesis, citrate is used as both reducing agent and capping agent; however, citrate is not a strong enough reducing agent to rapidly generate atomic gold at room temperature; therefore, the synthesis is carried out at elevated temperature, typically boiling. The AuNPs’ size is controlled predominantly by the ratio of citrate:Au, where more citrate results in more rapid nucleation and, subsequently, smaller particles. Particle size and distribution may also be controlled by pH [67], temperature [68], and order of reagent addition [69]. While AuNPs with an average diameter of 4 nm have been reportedly synthesized by the Turkevich method with minor modifications [70], it is far more common for particles to be larger than 10 nm in diameter.

**Sodium borohydride** (Figure 3B) is often implemented as a strong reducing agent in the synthesis of AuNPs enabling the reaction to be performed at room temperature and allowing for rapid nucleation and formation of smaller AuNPs, frequently sub 5 nm. Like the Turkevich method, citrate may be included; however, when NaBH_4_ is used as a reducing agent, citrate severs solely as a capping agent [71]. Alternatively, citrate can be replaced by other hydrophilic capping agents such as alginate [72] or chitosan [73]. The synthesis may also be performed in non-polar solvents such as chloroform, implementing hydrophobic capping agents such as CTAB [74] and ODA [75,76]. Finally, capping agents may be omitted entirely to produce “bare” AuNPs [77].

**Brust–Schiffrin** (Figure 3C) synthesis is a two-phase approach to produce alkanethiol-capped AuNPs which are soluble in hydrophobic solvents. TOAB is employed to transfer AuCl_4_^-^ from the aqueous phase to an organic phase, typically toluene; NaBH_4_ is used to reduce the gold salt in the presence of a capping agent, traditionally an alkanethiol. The capping agent first used, and still commonly used today, is the alkanethiol dodecanethiol [78]; however, this may be replaced with other alkanethiols such as pentanethiol [79] or hexanethiol [80], surfactants such as CTAB or CTAC [81], or even ionizable molecules for the synthesis of water soluble AuNPs, for example via the use of MPA [82].

**Seeded growth** (Figure 3D) is synthetic process of first producing Au^0^ clusters, often via NaBH_4_, although Turkevich/Frens particles may also be used as seeds, which are then introduced as presynthesized nuclei into a growth solution. Essentially, the particle number concentration of the resulting solution can be finely controlled by varying the number of nuclei introduced and the final particle size is regulated by the gold concentration in the growth solution. This approach is not particularly well suited to the formation of ultrasmall AuNPs and is more commonly employed for the preparation of particles over a large size range [83,84]. As well as being applicable over large size ranges, seeded growth is also capable of producing a wide variety of shapes by using different shape directing agents, such as CTAC for spheres [85] and cubes [86], CTAC/NaI for triangles [87], CTAB/CTAC/HQL for bipyramids/javelins [88], PVP for stars [89], and BDAC/CTAB for rods [90].

The methods outlined in the previous sections are summarized in Table 2.

## 4. Ultrasmall-in-Nano—Approaches and Applications

One approach which has been gaining popularity in the past few years is referred to as an ultrasmall-in-nano [30] construct: ‘ultrasmall’ represents the sub-5 nm gold cores, and ‘nano’ represents the 100–500 nm clusters of said AuNPs. While gold itself is inert and biocompatible, problems arise when considering persistence of the gold in the body [30]. The body’s main mechanism for excretion of compounds from circulation, specifically, via the renal pathway, is not efficient at removing particles larger than 5 nm in diameter [100]. This is mainly due to the functional pore size of the glomerular capillary wall of 4.5–5 nm [101] and larger particles will instead rely on elimination via the hepatobiliary pathway. Consequently, particles below this threshold are desirable to avoid the potential retention of gold in the body. However, several desirable properties result from particles only in the 100–500 nm range. These properties include increased circulation times [102], superior accumulation in tumor tissue due to the enhanced permeability and retention effect [103], and specifically in the case of AuNPs strong absorbance in the phototherapeutic window [104]. A work around for these diametrically opposed concepts is to have a system which can be converted from the latter to the former after its function has been served, i.e., from larger (100–500 nm) nanoparticles to ultrasmall (sub-5 nm) particles.

To display the desired optical properties, specifically, a bathochromic shift in absorbance upon clustering, the ultrasmall particles must come into proximity of one another. The effect of interparticle electromagnetic coupling is proportional to the inverse of interparticle distance [105], suggesting that the most prominent bathochromic shift is obtained when the ultrasmall particles are as close as possible without touching to avoid irreversible fusion, as would be observed with other aggregation methods, for example, upon the addition of glucosamine phosphate to small AuNPs [106].

Many approaches have been reported for forming ultrasmall-in-nano constructs; while they all follow what is essentially the same schematic, depicted in Figure 4, and have some common characteristics, they differ in the approach used to cluster the ultrasmalls and in the application of the nano construct. Some of these approaches are outlined in the following sections and summarized in Table 3.

### 4.1. Approaches to Clustering

#### 4.1.1. Small Molecule Crosslinking

Mellor et al. [75] showed that using a labile dithiol molecule—ethylene glycol bis-mercaptoacetate (EGBMA)—they were able cluster octadecylamine (ODA) AuNPs (sub-5 nm) and demonstrated that, under physiological conditions, the clusters can revert to ultrasmall AuNPs. The nano constructs exhibited strong absorbance in the phototherapeutic window and were labelled with a Raman reporter, allowing for specific detection in a biological matrix, demonstrating its feasibility for future theranostic applications.

#### 4.1.2. Coating of Liposomes

Rengan et al. [107] prepared a formulation that can be described as ultrasmall-on-nano, by first forming 1,2-distearoyl-sn-glycero-3-phosphocholine (DSPC): cholesterol liposomes in the nano range, and subsequently coating them with ultrasmall AuNPs (2–8 nm). They demonstrate the ability of these particles, following intratumoral injection, to kill cancer cells using PTT with complete ablation of the tumor mass following 750 nm laser illumination. The particles are shown to be degraded in hepatocytes and cleared through the hepato-biliary and renal routes. On days 1, 7, and 14, the %ID detected in the liver was 52%, 9.8%, and 3%, respectively, and in the kidney was 2.7%, 0.25%, and 0.22%, respectively. This demonstrated a significant reduction in gold levels in just 14 days, and the authors hypothesized that renal excretion would be increased when the constructs are subjected to both photothermal and enzymatic degradation.

#### 4.1.3. DNA Assembly

Chou et al. [108] designed a core-satellite architecture whereby AuNPs (13 nm cores, and 3 or 5 nm satellites) coated with thiolated, single-stranded DNA were then assembled using complementary sequence linker DNA. They demonstrated that, by careful selection of the DNA sequence and capping ligand, they could tune a number of properties: the constructs’ ability to encapsulate small molecules, their propensity for cellular uptake, elimination, and tumor-targeting. In addition, in this study, gold content in urine was quantified 48 h after systemic injection of cores, satellites, and core-satellite constructs. They found the highest levels of renal excretion for the smallest particles tested—15 %ID for 3 nm satellites with a 1 kDa PEG coating—and decreasing levels with increasing particle size or PEG molecular weight. Similarly, for the core-satellite constructs, they found that the urine levels were proportional to the size of the satellite tested, and lower than that of the respective satellite alone. This finding suggests, as would be expected, that only the satellites are being excreted and not the 13 nm cores.

#### 4.1.4. Encapsulation/Ionic Interaction

Voliani and their team demonstrated, in multiple articles [109,110,111,112,113,114], their ultrasmall-in-nano construct, referred to as ‘passion fruit’-like, formed by ionic interaction between poly(l-lysine) (PL) and poly(sodium 4-styrene sulfonate) (PSS)-coated ultrasmall AuNPs (~3 nm). They studied their tumor-targeting ability following modification with a peptide [109]; biocompatibility and excretion in murine models where they showed excretion from both renal and biliary pathways [110]; pharmacokinetics following inhalation [111], showing accumulation in the lungs, translocation to secondary organs, and almost complete excretion within 10 days; and the suitability of their ultrasmall-in-nano construct for PTT applications [114]. The researchers monitored the biodistribution and excretion for 10 days following intravenous injection through the tail vein. They observed that the gold concentration in the liver decreased over the course of time, and that gold was detected in both urine and faces for the duration of the experiment for a cumulative excretion of ~16 %ID in 10 days.

Higbee-Dempsey et al. [115] prepared ultrasmall p-MBA-AuNPs coated with thiolated dextran (2.1 nm); the dextran was made to be hydrophobic by the covalent incorporation of acetyl groups. The hydrophobic AcetalDextran-pMBA-AuNPs were mixed with poly(ethylene glycol)-block-poly(ε-caprolactone) (PEG-PCL), resulting in micellization and dense packing of the ultrasmall cores. Acetyl groups were shown to be cleaved in an acidic environment, resulting in a hydrophilic polymer, demicellization, and dispersion of gold cores. The group showed clearance of gold from mice organs over 3 months, following a single bolus injection via the tail vein. They report that levels in the liver and spleen reduced by 86% and 72%, respectively, and that gold is detected in urine and faeces at decreasing levels over the time course; however, the reported levels represent samples collected on the day of sacrifice, and do not represent the cumulative excretions between time points.

Cheheltani et al. [116] encapsulated ultrasmall GSH-coated AuNPs (2–5 nm) in a biodegradable poly di(carboxylatophenoxy)phosphazene (PCPP) polymer; they were able to control the size of the particles, and therefore, the position of the SPR peak in the NIR region, by varying the amount of polyethylene glycol-polylysine block co-polymer in the formulation. They demonstrate the potential of these constructs as computed tomography (CT) and photoacoustic (PA) imaging contrast agents both in vitro and in vivo. They also demonstrate the ability of the clusters to degrade upon incubation in serum.

Yahia-Ammar et al. [117] first synthesized ultrasmall GSH-coated AuNPs (~2 nm); these particles were then encapsulated by the addition of poly(allyl amine hydrochloride) (PAA HCl) polymer. They show that the clustered particles exhibit enhanced fluorescence compared to the ultrasmall particles, with quantum yield increasing from 7 to 25%. They also demonstrated the increased cellular uptake of these particles and the application of this to enhanced cellular delivery of both peptides and antibodies.

Tam et al. [118] clustered ultrasmall lysine/citrate-capped AuNPs (4.1 nm) via ionic interaction with biodegradable triblock copolymer of polylactic acid and polyethylene glycol (PLA(2K)-PEG(10K)-PLA(2K)). Upon cluster formation, the absorbance maximum shifted from 520 nm into the NIR region with a fairly constant absorbance of 700 to 900 nm. Limited degradation was observed after 4 weeks at neutral pH, whereas almost total degradation was observed after just 1 week at pH 5, a finding attributed to the stability of PLA at neutral pH. Nanocluster degradation was confirmed in vitro by TEM and scattering spectra from hyperspectral images of treated and untreated murine macrophage cells over 168 h.

Deng et al. [119] worked with 6.1 nm particles, which is on the upper limit of what may be considered ultrasmall, as acknowledged by the researchers; however, the approach may be transferable to smaller particles to ensure excretability. Their system is based on the self-assembly of AuNPs within a novel comb-like amphipathic polymer composed of hydrophobic poly(ε-caprolactone)/poly(2-hydroxyethyl methacrylate) (PCL-PHEMA) and hydrophilic poly(2-(2-methoxyethoxy) ethyl methacrylate) (PMEO_2_MA). The group tested the potential of the particles for PTT by exposing solutions of various concentrations to 808 nm NIR laser at a power density of 1.5 W cm^−2^; at the highest concentration (0.4 mg mL^−1^), they observed heating up to 71 °C after laser irradiation for 5 min. They also loaded DOX into the particles, which was released following laser irradiation to demonstrate the combined chemotherapeutic and phototherapeutic abilities of the construct. The constructs were shown to be suitable as contrast agents for both CT and PAT imaging. The researchers monitored the levels of the gold in the tumor, vital organs (heart, liver, spleen, lung, and kidney), and vital metabolic products (bile, urine, and faces) for 7 days following intratumoral injection. They found that levels in the tumor decreased over the course of time; the levels in the vital organs increased until day 2, but then had decreased by day 7, and the levels in metabolic products increased over the course of time. The increasing renal excretion over the course of the experiment, with 4 %ID detected in the urine on day 7, suggests disassembly of the constructs back to ultrasmall AuNPs, and that 6 nm AuNPs are still renally excreted, if only to a lower extent than smaller particles.

### 4.2. Accomplishments of Ultrasmall-in-Nano Constructs

It is clear that ultrasmall-in-nano provides a viable route to produce constructs with the properties of particles larger than their constituent cores; in various studies, they have been shown to have absorbance in the phototherapeutic window [75,107,116,118,119], to be SERS active [75], to be suitable as PTT agents [107,114,119], and to function as PA and CT contrast agents [116], amongst other key characteristics.

What has also been demonstrated, to a lesser extent, is the excretion of ultrasmall particles following administration of nano constructs. Where in vivo studies have been performed to monitor retention and/or excretion, they have demonstrated quite varied results, suggesting that the excretion rate is highly dependent on the approach to clustering, as well as the size of the ultrasmall particle. Rengan et al. [103] showed fairly quick decrease in levels of gold detected in the liver of 52% to 3%ID over 14 days, whereas Higbee-Dempsey et al. [115] reported ~85% elimination from the liver over three months. It is promising that studies such as those performed by Chou et al. [108] demonstrated that the rate of excretion of ultrasmall-in-nano constructs is proportional to the sizes of the constituent ultrasmall particles. This highlights the necessity for the entirety of the construct to be comprised of ultrasmall particles, as the 13 nm cores do not appear to have been excreted.

Several studies, where excretion has not yet been demonstrated experimentally, have reported that their ultrasmall-in-nano constructs are able to liberate ultrasmall particles [75,117,119]. This, combined with the data of the aforementioned in vivo studies, suggests that there are several emerging constructs with the potential to deliver the capabilities of nanoparticles, while retaining the excretable nature of ultrasmall particles.

## 5. Conclusions and Future Perspectives

Optical properties, opsonization, cellular internalization, renal clearance, biodistribution, tumor accumulation, and toxicity all display size-dependent relationships with gold nanoparticles. In some regards, smaller particles are favored, for example for increased renal clearance; in others its larger particles, such as to display strong absorbance in the phototherapeutic window. Clearly, it is not possible to have a single static construct which displays all these properties; however, ultrasmall-in-nano promises a dynamic structure to provide the advantages of the small and the large.

Where the properties of the larger particles are disadvantageous, work-arounds have been proposed which usually entail the modification of the particles’ surface, for example, PEGylation to reduce the propensity for opsonization, or conjugation to a targeting moiety to increase tumor accumulation.

It is noteworthy that the examples of ultrasmall-in-nano reviewed here were all from the last decade or so; this emphasizes how novel this work is and the interest its subject matter has gained in a relatively short time.

Ultrasmall-in-nano constructs have already been shown to deliver a lot of what the concept promises, from phototherapeutic absorbance and SERS activity to declustering and excretion. Future works will likely focus on combining many of these properties into a single construct and demonstrating complete excretion in a reasonable timeframe.

## Figures and Tables

**Figure 1 nanomaterials-12-02476-f001:**
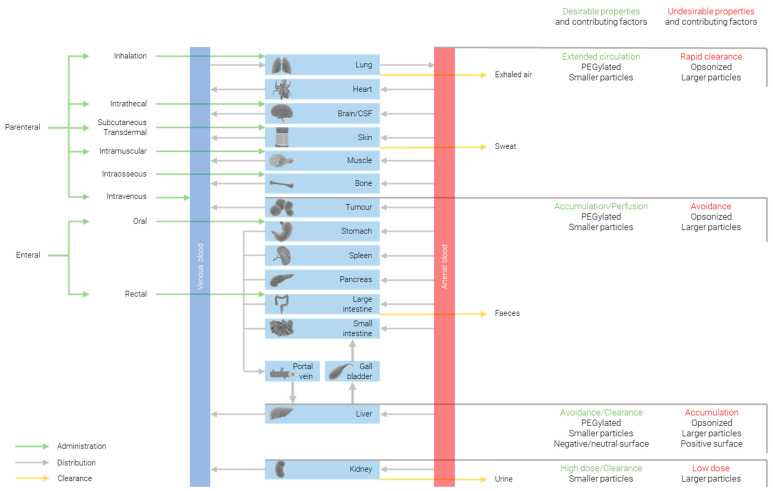
Approaches for modulating biodistribution of nanoparticles.

**Figure 2 nanomaterials-12-02476-f002:**
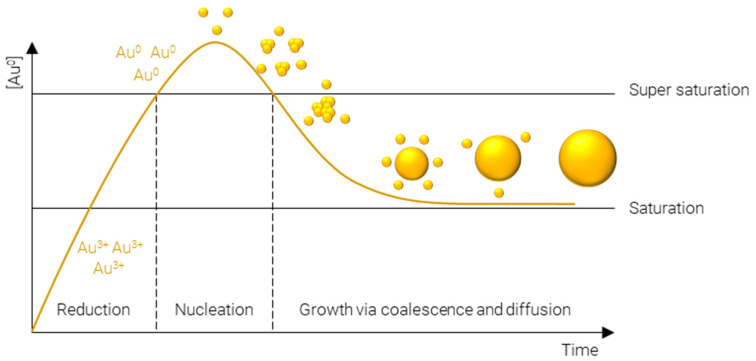
LaMer model of metal nanoparticle formation.

**Figure 3 nanomaterials-12-02476-f003:**
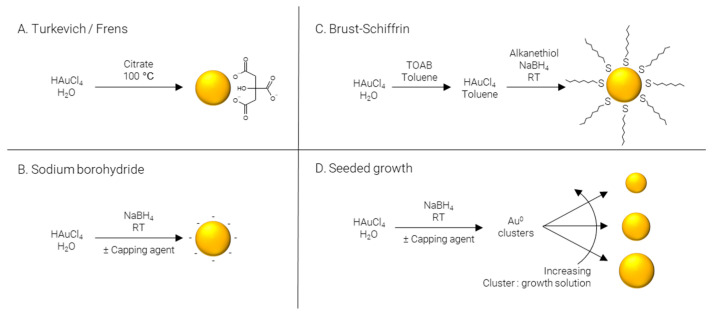
Methods of AuNP synthesis.

**Figure 4 nanomaterials-12-02476-f004:**
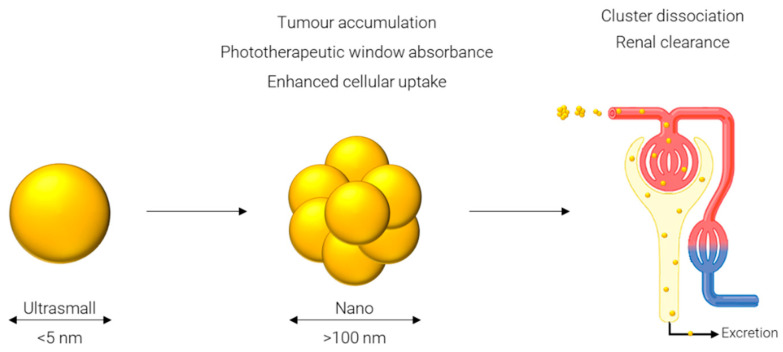
Generalized principle of the ultrasmall-in-nano approach.

**Table 1 nanomaterials-12-02476-t001:** Abbreviations for reagents use in AuNP synthesis.

Abbreviation	Meaning
BDAC	Benzyldimethylhexadecylammonium chloride
CTAB	Cetyltrimethylammonium bromide
CTAC	Cetyltrimethylammonium chloride
GSH	Glutathione
HAuCl_4_	Chloroauric acid
HQL	8-hydroxyquinoline
MPA	Mercaptopropionic acid
NaBH_4_	Sodium borohydride
NaI	Sodium iodide
ODA	Octadecylamine
PVP	Polyvinylpyrrolidone
TOAB	Tetraoctylammonium bromide

**Table 2 nanomaterials-12-02476-t002:** Methods of AuNP Synthesis. Focusing mainly on ultrasmall spheres, with several prominent examples of methods for synthesizing larger or non-spherical particles.

Method of Synthesis	Size Range	Shape	Surface Chemistry	Polarity	Solvent	Ref.
Turkevich	15–24 nm	Sphere	Citrate	Hydrophilic	H_2_O	[65]
Frens	16–147 nm	Sphere	Citrate	Hydrophilic	H_2_O	[66]
Turkevich/Frens	4 nm	Sphere	Citrate	Hydrophilic	H_2_O	[70]
Sodium borohydride	3–5 nm	Sphere	Citrate	Hydrophilic	H_2_O	[71]
Sodium borohydride	3.3–12	Sphere	Alginate	Hydrophilic	H_2_O	[72]
Sodium borohydride	3.5–14 nm	Sphere	Chitosan	Hydrophilic	H_2_O	[73]
Sodium borohydride	3–14 nm	Sphere	CTAB	Hydrophobic	CHCl_3_	[74]
Sodium borohydride	4.7 nm	Sphere	ODA	Hydrophobic	CHCl_3_	[75]
Sodium borohydride	3 nm	Sphere	ODA	Hydrophobic	CHCl_3_	[76]
Sodium borohydride	3–5 nm	Sphere	Bare	Hydrophilic	H_2_O	[77]
Turkevich/Frenz—modified	3.6–13 nm	Sphere	Citrate/tannic acid	Hydrophilic	H_2_O	[91]
Turkevich/Frenz—modified	3.5–15 nm	Sphere	PDEAEM	Hydrophilic	H_2_O	[92]
Turkevich/Frenz—modified	2–330 nm	Sphere	Citrate	Hydrophilic	H_2_O	[93]
Brust-Schiffrin	1–3 nm	Sphere	Dodecanethiol	Hydrophobic	Toluene	[78]
Brust-Schiffrin	5 nm	Sphere	Pentanethiol	Hydrophobic	Toluene	[79]
Brust-Schiffrin	2 nm	Sphere	Hexanethiol	Hydrophobic	Toluene	[80]
Brust-Schiffrin	10 nm	Sphere	CTAB/CTAC	Hydrophobic	Toluene	[81]
Brust-Schiffrin	3 nm	Sphere	MPA	Variable	Toluene/H_2_O	[82]
Seeded growth	8.4–180.5 nm	Sphere	Citrate	Hydrophilic	H_2_O	[83]
Seeded growth	15–300 nm	Sphere	Citrate	Hydrophilic	H_2_O	[84]
Seeded growth	5–150	Sphere	CTAC	Hydrophilic	H_2_O	[85]
Seeded growth	60 nm	Triangle	CTAC/NaI	Hydrophilic	H_2_O	[87]
Seeded growth	76 nm	Cube	CTAC	Hydrophilic	H_2_O	[86]
Seeded growth	40–300 nm	Bipyramid/Javelin	CTAB/CTAC/HQL	Hydrophilic	H_2_O	[88]
Seeded growth	45–116 nm	Star	PVP	Hydrophilic	DMF	[89]
Seeded growth	10–100 nm	Rod	BDAC/CTAB	Hydrophilic	H_2_O	[90]
Other—GSH reduction	2.5 nm	Sphere	GSH	Hydrophilic	H_2_O	[94]
Other—GSH reduction	2.3 nm	Sphere	GSH/cysteamine	Hydrophilic	H_2_O	[95]
Other—HEPES reduction	23 nm	Star	HEPES	Hydrophilic	H_2_O	[96]
Other—TBAB reduction	2–7 nm	Sphere	Oleylamine	Hydrophobic	DCM	[97]
Other—TBAB reduction	3–10 nm	Sphere	Oleylamine	Hydrophobic	Hexane	[98]
Other—thermal reduction	2 nm	Sphere	PEG	Hydrophilic	H_2_O	[99]
Other—mechanochemical	1–4 nm	Sphere	Various	Various	None	[64]

**Table 3 nanomaterials-12-02476-t003:** Methods of generating ultrasmall-in-nano constructs.

Ultrasmall (Surface Chemistry, and Size)	Nano (Clustering Principle, and Size)	SPR	Reversible	Refs.
ODA 4.67 ± 1.74 nm	Crosslinking with EGBMA 254–278 nm	710 nm	Yes	[75]
NA 2–8 nm	Coating of DSPC: cholesterol liposomes 100–120 nm	760 nm	Yes	[107]
Tannic acid and/or citrate 3, 5, and 13 nm	Single stranded DNA-coated AuNPs + complementary linker 50–150 nm	Nr	Yes	[108]
PSS ~3 nm (varies with article)	Ionic interactions with PL ~100 nm (varies with article)	530 nm	Yes	[109,110,111,112,113,114]
AcetalDextran-pMBA-AuNPs 2.1 ± 0.5 nm	Encapsulation in PEG-PCL 111.1 ± 38 nm	Nr	Nr	[115]
11-MUA or GSH 2–5 nm	Encapsulation in PCPP 40–500 nm	>650 nm	Yes	[116]
GSH ~2 nm	Encapsulation in PAA HCl 120 nm	Nr	Nr	[117]
Citrate/lysine 4.1 ± 0.8 nm	Interaction with PLA(2K)-PEG(10K)-PLA(2K) 83.0 ± 4.6 nm	Broad, NIR absorbance	Yes	[118]
NA 6.1 ± 1.8 nm	Self-assembly with PCL-PHEMA and PMEO_2_MA 300 nm	800 nm	Nr	[119]

Nr: Nor reported.

## Data Availability

Not applicable.

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
