# Peer review of "Ultrasmall-in-Nano: Why Size Matters"

_nanomaterials, 2022, doi:10.3390/nano12142476_

Round 1

Reviewer 1 Report

The manuscript entitled “Ultrasmall-in Nano:Why Size Matters” by Ryan D. Mellor and Ijeoma F. Uchegbu presents a the impact of size (in nano scale) on different properties and behavior of nanoparticles and their applications. I found very interesting the paragraph in rows 434-438, which is however, somehow lost in the text. I would recommend emphasizing this observation more, because from the point of view of future applications this is probably a very promising path – to let bigger particle work and then shrink it down to the ultra-small one and eliminate it from the body by renal system.

Author Response

The manuscript entitled “Ultrasmall-in Nano:Why Size Matters” by Ryan D. Mellor and Ijeoma F. Uchegbu presents a the impact of size (in nano scale) on different properties and behavior of nanoparticles and their applications. I found very interesting the paragraph in rows 434-438, which is however, somehow lost in the text. I would recommend emphasizing this observation more, because from the point of view of future applications this is probably a very promising path – to let bigger particle work and then shrink it down to the ultra-small one and eliminate it from the body by renal system.

We thank reviewer 1 for their feedback, we are pleased that the potential of the ultrasmall-in-nano concept is being acknowledged. This point is expanded upon in “Conclusions and Future Perspectives“

Reviewer 2 Report

Manuscript presents a review of examples of fabrication and medical applications of ultrasmall gold nanoparticles published for the last decade. The review will be useful for specialists working in the field of applications of surface plasmon resonance (SPR) in biomedicine. Paper can be accepted for publication after minor revision considering the following comments:

1.      Paper includes 4 Tables for which there are no references in the text. For example, the abbreviations for most of chemical reagents are given in Table 2, but this is not evident for the readers until they will reach page 7 of the paper.

2.      Page 2, line 135 – “Meanwhile, smaller (<5 nm) show higher rates ><5nm) show higher” must be replaced with “Meanwhile, smaller ones (<5 nm) show higher rates (<5nm) show higher”.

3.      Page 5, lines 204 and 206 – abbreviations for BALB and MTT must be introduced.

4.      Page 7, line 310 – “Figure 4” must be replaced with “Figure 3”, there is no figure 4 in the paper.

5.      Page 13, Table 4 – it looks like “Nr” is stands for “No response”, however this must be clearly mentioned under the Table or in the text along with reference to the Table 4.

6.      Page 13 – it is not clear if paragraph “Accumplishments of ultrasmall-in-nano constructs” is part of Section 4 (like 4.2, for example) or this is a new Section?  In addition, there is a misprint in this title, and  Accumplishments” must be replaced as   Accomplishments”.

7.      From my point of view it would be useful to add a reference to a very recently published review -  Langer, J.; Jimenez de Aberasturi, D.; Aizpurua, J.; Alvarez-Puebla, R.A.; Auguié, B.; Baumberg, J.J.; Bazan, G.C.; Bell, S.E.; Boisen, A.; Brolo, A.G.; et al. Present and Future of Surface-Enhanced Raman Scattering. ACS Nano (2020) 14(1) 28-117.

8.      Page 14, list of references - for references 9, 11 and 13 “DOI” is absent.

Author Response

Manuscript presents a review of examples of fabrication and medical applications of ultrasmall gold nanoparticles published for the last decade. The review will be useful for specialists working in the field of applications of surface plasmon resonance (SPR) in biomedicine. Paper can be accepted for publication after minor revision considering the following comments:

  1. Paper includes 4 Tables for which there are no references in the text. For example, the abbreviations for most of chemical reagents are given in Table 2, but this is not evident for the readers until they will reach page 7 of the paper.

Missing references to the tables have been added to the text.

  1. Page 2, line 135 – “Meanwhile, smaller (<5 nm) show higher rates ><5nm) show higher” must be replaced with “Meanwhile, smaller ones (<5 nm) show higher rates (<5nm) show higher”.

Corrected.

  1. Page 5, lines 204 and 206 – abbreviations for BALB and MTT must be introduced.

The chemical identity of MTT has been added in the text.

The definition of BALB/c is omitted as it is merely the name of a mouse strain and does not convey much scientific meaning.

  1. Page 7, line 310 – “Figure 4” must be replaced with “Figure 3”, there is no figure 4 in the paper.

Corrected.

  1. Page 13, Table 4 – it looks like “Nr” is stands for “No response”, however this must be clearly mentioned under the Table or in the text along with reference to the Table 4.

Nr in this context stands for “not reported” and has been added to the table footer.

  1. Page 13 – it is not clear if paragraph “Accumplishments of ultrasmall-in-nano constructs” is part of Section 4 (like 4.2, for example) or this is a new Section?  In addition, there is a misprint in this title, and  “Accumplishments” must be replaced as   “Accomplishments”.

Corrected both typo and missing numbering.

  1. From my point of view it would be useful to add a reference to a very recently published review -  Langer, J.; Jimenez de Aberasturi, D.; Aizpurua, J.; Alvarez-Puebla, R.A.; Auguié, B.; Baumberg, J.J.; Bazan, G.C.; Bell, S.E.; Boisen, A.; Brolo, A.G.; et al. Present and Future of Surface-Enhanced Raman Scattering. ACS Nano (2020) 14(1) 28-117.

Reference added to “Effect on optical properties”

  1. Page 14, list of references - for references 9, 11 and 13 “DOI” is absent.

Missing DOIs have been added.

We thank reviewer 2 for their feedback, their attention to detail evident, and the points raised have been addressed in the manuscript.

Reviewer 3 Report

The manuscript is properly prepared, from where a good image on the subject may be done. The authors describe clearly and detailed the applications of AuNPs. However, I should like to find several lines concerning the labeled compositions that are used as radiopharmaceutics.

Otherwise, I agree the publication of this study.

Author Response

The manuscript is properly prepared, from where a good image on the subject may be done. The authors describe clearly and detailed the applications of AuNPs. However, I should like to find several lines concerning the labeled compositions that are used as radiopharmaceutics.

Radiopharmaceuticals were omitted from this review as no literature was found to suggest that there is a link between this application and the size of the particle.

Otherwise, I agree the publication of this study.

We thank reviewer 3 for their feedback, and we are pleased that they find the review to be sufficiently clear and detailed.